# Testing the reliability and validity of a newly graduated nurses' teaching experience scale

Tomoki Doi[1,2]*, Yasuko Shimizu[3]

1 Faculty of Nursing, Osaka Medical and Pharmaceutical University, Takatsuki, Osaka, Japan, 2 Division of Health Sciences, Osaka University Graduate School of Medicine, Suita, Osaka, Japan, 3 Division of Health Sciences, Osaka University Graduate School of Medicine, Suita, Osaka, Japan

* tomoki.doi@ompu.ac.jp

## Abstract

Guideline-based education provided to newly graduated nurses (NGNs) involves nurse instructors who may be inexperienced, leading to increased burden and anxiety related to teaching NGNs. To support these nurses, this study aimed to test the reliability and validity of a scale measuring the teaching experiences of NGNs. A cross-sectional survey of the original draft of the scale consisting of 72 items, developed based on previous research, was conducted among nurses in their third to fifth years of working in Japanese hospitals with 400 or more beds. Exploratory factor analysis and factor-item consistency were assessed. Responses were obtained from 163 participants, and data from 157 participants were included in the analysis. Exploratory factor analysis revealed 27 items comprising five factors (Factor 1: Experience of receiving cooperation from others, Factor 2: Experience of being there for the NGNs, Factor 3: Experience of feeling positive about teaching, Factor 4: Experience of struggling with teaching, and Factor 5: Experience of teaching according to level of achievement). Cronbach's alpha coefficients ranged from .849 to .923 for each factor and Cronbach's alpha was .924 overall. Factor-item consistency revealed the following: $\chi^2(314) = 637.117$ (p < .001), CFI = .871, RMSEA = .081. Each factor in the scale reflects the characteristics of the experience of teaching NGNs, and was found to be reliable and valid. This scale will help managers understand the teaching experiences of novice nurses and provide support based on individualized learning experiences.

## Introduction

In Japan, in-service education is provided to newly graduated nurses (NGNs) who have recently joined a hospital based on the Guidelines for Training of New Nursing Staff [Revised Edition] [1]. These guidelines direct the training of NGNs in the knowledge and skills necessary to provide safe, high-quality nursing care and support the development of all staff members through their guidance of NGNs. The guidelines

**Data availability statement:** The data used in this analysis is owned by the researchers but can be provided upon request. While the data was obtained from subjects who consented to its use solely for this research purpose, consent was not obtained for the public release of individual collected data. Furthermore, anonymization was completed after the dataset was created, making it difficult to retrospectively obtain consent from research subjects for uploading the data to a repository or similar platform. For these ethical reasons, the data cannot be provided for other purposes. These restrictions were imposed by the Ethical Review Board of Osaka University Hospital (Email: rinri@hp-crc.med.osaka-u.ac.jp).

**Funding:** The author(s) received no specific funding for this work.

**Competing interests:** The authors have declared that no competing interests exist.

indicate the need not only to assign a practitioner to provide practical guidance and evaluation of clinical practice to NGNs but also to establish an organizational structure in which all department staff members provide multiple layers of support to the NGNs. In addition, a system has been established in which the entire staff provides guidance to NGNs, ensuring that the burden of providing guidance is not placed solely on the supervisor [2]. In Japan, nurses with a minimum of 2.4 years of practical nursing experience and an average of five years of practical nursing experience serve as field supervisors [3], but nurses who lack sufficient practical nursing experience are still involved in teaching NGNs.

Because nurses work in shifts, the workday of NGNs may not coincide with the workday of their on-the-job supervisors. Even in such cases, nurses are still needed to supervise the NGNs. In such cases, nurses who are not on-the-job supervisors or who have little experience must be involved in teaching NGNs. Nurses who have not yet been assigned the role of a field instructor differ from field instructors in that they are expected to perform the role of a field instructor [2,4–7], communicate [5,6], manage goals and evaluate stress [8], and plan in-service guidance [7,9–12]. In addition, even if nurses are acting as instructors for the first time, it can be said that they are teaching NGNs in an unfamiliar situation, despite having had the opportunity to learn the necessary material. Therefore, teaching NGNs can be a burden for nurses who are not instructors or who have fewer years of experience. In terms of instructors, they reported challenges related to their responsibility for the failed growth of NGNs [5] and expressed feeling burdened by the instructor role itself [5,13,14]. Additionally, they experienced a lack of support for the instructor role [15]. Thus, nurses who are not hands-on supervisors or who have fewer years of experience may experience more difficulty and anxiety about teaching NGNs.

However, at this stage, nurses are busy with their daily tasks and have limited ability to recognize and articulate their additional challenges related to teaching NGNs. Therefore, creating a simple scale to capture the experience of teaching NGNs and quantifying the aspects of teaching them, can aid in providing support based on nurses' individual teaching experiences.

Instructors grow by reflecting on their experiences of failure during teaching, seeking individualized teaching methods [16], and viewing teaching NGNs as an opportunity for their own growth [14,17]. In addition, new instructors among the nurses felt they lacked the skills to be effective mentors and had difficulties controlling their emotions and collaborating with their seniors. However, they experienced joy in teaching and noticed positive changes in themselves [7]. Thus, many previous studies have clarified the experiences that nurses have had in teaching NGNs from a qualitative perspective. The only study that could quantify the experience of teaching NGNs was the Self-evaluation Scale of Preceptor Role Performance for New Graduate Nurses [18], which was developed in Japan. However, this scale was created from the perspective of evaluating the achievement of the instructor role, which differs from our scale, which was designed to clarify the experience of teaching NGNs.

As learning from teaching NGNs affects a nurse's ability to practice nursing [19], the experience of teaching NGNs may positively impact these nurses. Therefore, it

is necessary to consider support for teaching NGNs, even for nurses who lack sufficient practical and teaching experience, and to clarify their experiences of teaching NGNs. Therefore, this study focused on nurses who had not served as in-service supervisors and those with a few years of experience to develop a scale that measures their experiences in teaching NGNs, and tested its reliability and validity.

### Terminology

**Experience of teaching NGNs.** Nurses' experiences of teaching NGNs in a hospital setting included both direct contact with NGNs and the experience of seeking and receiving support to facilitate their teaching, such as requesting cooperation from supervisors, colleagues, and other nurses.

**Nurses with less experience in teaching NGNs.** This refers to nurses working in Japanese hospitals who had never served as field supervisors to the fiscal year at the time of the survey, as well as nurses who had been assigned a field supervisor role for approximately six months during the fiscal year at the time of the survey. Nurses who were assigned new teaching positions before the fiscal year at the time of the survey were excluded. In Japan, the new fiscal year begins in April, and new recruits enter the workforce at that time.

### Purpose

We developed and tested the reliability and validity of a scale that quantitatively measures the teaching experiences of nurses with less experience in teaching new nurses. The developed scale reflected the characteristics of the experience of teaching NGNs and was found to be reliable and valid.

## Materials and methods

### Study design

We employed a cross-sectional design.

### Participants

Nurses with three to five years of experience working in Japanese hospitals with more than 400 beds who agreed to participate in this study were included. Nurses with these years of experience were targeted based on Fujishiro's survey [3], which reported that the minimum number of years of experience for teaching NGNs was 2.4 years, and the average number of years was five. As this study aimed to develop a scale to measure the experiences of nurses with less teaching experience with NGNs, third- to fifth-year nurses were targeted. Nurses who had previously taught NGNs were excluded. Hospitals with more than 400 beds were selected because 94.4% of new employees are hired at hospitals with more than 400 beds, compared to 52.9% at hospitals with fewer than 400 beds [20].

### Instrument

**Draft scale.** We independently drafted questionnaire items exploring the experiences of teaching NGNs based on a literature review. Consequently, 222 specific experience items were extracted, and these experiences were qualitatively aggregated based on similarities in content and expression under the supervision of a qualitative research expert. The final draft scale consisted of 72 items, and a pretest was conducted with three nurses who had at least one year of experience in teaching NGNs to confirm the clarity of the questionnaire and the ease of answering it. Participants responded to each item of the draft scale using a 4-point Likert scale (1 = never experienced, 2 = rarely experienced, 3 = often experienced, and 4 = very often experienced).

**Nursing competence.** There is a significant positive correlation ($r = .54$) between scores on the Self-Evaluation Scale of Preceptor Role Performance for New Graduate Nurses and nursing competence [21]. This study aimed to measure

the experiences of nurses with limited years of experience in teaching NGNs. However, the Self-Evaluation Scale of Preceptor Role Performance for New Graduate Nurses [20] was developed based on surveys of nurses who had more years of experience than the nurses targeted in this study, as well as those with more orientation experience. Therefore, we considered it unlikely to adequately reflect the nature of the experiences among the nurses this study focused on. Furthermore, since nurses with a certain level of nursing practice competence, even if they have limited years of experience, are assigned to teaching NGNs, this study examined the correlation with nursing practice competence rather than using the Self-Evaluation Scale of Preceptor Role Performance for New Graduate Nurses. The Clinical Nursing Competence Self-Assessment Scale (CNCSS) was used to assess nursing competence. This scale was developed by Maruyama et al. [22] and enables self-assessment of nursing practice competence from two perspectives: "frequency of implementation" and "degree of achievement." In this study, the "degree of achievement" score was used to analyze nursing competence, exploring it from the perspective of how confident the nurse is in their abilities, rather than the amount of nursing practice they perform. This scale has been validated for concurrent validity with a Japanese version of the 6-Dimension Scale of Nursing Performance developed in the United States [23]. This scale consists of three categories, 13 subcategories, and 64 items. A higher score indicates a greater ability to practice, and the Cronbach's alpha coefficient for the entire scale was.97 [23]. This scale was used because it measures nursing practice skills that are more in line with current trends than the Japanese version of the 6-Dimension Scale of Nursing Performance, as it includes concepts such as risk management and quality improvement, which are currently considered vital to medical and nursing practice.

**Sociodemographic characteristics.** Regarding sociodemographic characteristics, the participants were asked to indicate their sex, age, qualifications, highest level of education in nursing, years of nursing experience, years of experience in their current department, and whether they are in charge of an NGN.

**Data collection.** The survey was sent to hospital administrators who agreed with the study's purpose. These administrators distributed the paper-based survey to the target nurses at their respective hospitals. The survey form also included an explanation of the study and a question to confirm consent, after which participants were asked to return the survey to the researcher by mail. No specific location or time was designated for completing the questionnaire; participants were free to decide at their own discretion. Data were collected from October 10 to November 22, 2017.

**Data analysis.** After identifying and removing items that indicated ceiling and floor effects from the mean and standard deviation of the draft scale results, exploratory factor analysis was conducted. The factor extraction method used was the maximum likelihood method, the rotation method was promax rotation, the factor loading criterion was.500, and scale items were selected. The validity of the factor analysis results was verified using the Kaiser-Meyer-Olkin (KMO) test, Bartlett's sphericity test, and intraclass correlation coefficients. Cronbach's alpha coefficients were calculated to confirm the internal consistency of the factor analysis results. Furthermore, the research members involved in nursing education repeatedly discussed and confirmed the validity of each item. Then, the structural consistency of each factor and item was confirmed using $\chi^2$ values, CFI, and RMSEA. Finally, the correlation coefficient between the total score of all items obtained from exploratory factor analysis and the total score of the CNCSS was calculated. Data analysis was conducted using Microsoft Excel 2016, IBM SPSS version 25, and IBM AMOS version 25, with the significance level set at $\alpha = .05$.

**Ethical considerations.** Participants were informed in writing that participation in the study was voluntary, that there would be no disadvantages for not participating, and that anonymity would be maintained. Those who agreed to participate were asked to return the survey forms. This study was approved by the Ethical Review Board of Osaka University Hospital (No. 16500−2). The CNCSS was used with the developer's permission.

## Results

### Sociodemographic characteristics

Questionnaires were mailed to 241 participants in their third to fifth years at 35 facilities that had given prior consent to participate in the study; 163 returned the questionnaires, resulting in a response rate of 67.6%. Of these, data from

157 participants were included in the analysis (valid response rate: 65.1%), excluding six participants who had missing values of 20% or more. In total, 149 women (94.9%) and 8 men (5.1%) participated, with a mean age of 25.4 ± 3.3 years (Table 1).

## Exploratory factor analysis

The means and standard deviations for all 72 items were calculated to check for ceiling and floor effects. Since three items showed ceiling effects and two items showed floor effects, an exploratory factor analysis was conducted for all 67 items, excluding these five items.

After checking the initial scree plot and eigenvalues, exploratory factor analysis was conducted using the maximum likelihood method and Promax rotation, with the number of factors set to five. The criterion for factor loadings was set at.500. After deleting items with low factor loadings and organizing the items while checking the number of intra-class correlations, the final result consisted of 27 items across five factors. Item 8: "Nurses with little experience in teaching NGNs encouraged each other" had a factor loading of.477 for Factor 1, while Item 15—"I made time for the NGNs to tell me what they were worried about in their work"—had a factor loading of.487 for Factor 2, both slightly below.500. However, regarding item 8: "Nurses with little experience in teaching NGNs encouraged each other," it was determined that the presence or absence of peer support among nurses with limited experience in teaching NGNs is an important factor in advancing NGNs' training; thus, it was included in Factor 1. Additionally, regarding item 15—"I made time for the NGNs to tell me what they were worried about in their work"—it was deemed essential to carve out time to engage with NGNs, even amidst a busy schedule, which led to its inclusion in Factor 2. The factor loading of item 15: "I made time for NGNs to talk about their concerns at work" on Factor 2 was only.487, slightly lower than.500, but was included as it was considered an important item that constituted a factor. The correlation coefficients between each factor displayed weak correlations between Factor 4 and Factor 2, as well as between Factor 4 and Factor 3, with value of $r = .052$ and $r = .056$, respectively. However, all other factors showed moderate to high positive correlations ($r = .350$ to.650). The cumulative contribution ratio was 59.572%, the KMO measure of factor validity was.867, Bartlett's sphericity was significant at $p < .001$, and the results

**Table 1. Summary of the participants (n = 157).**

| Items | | n (%) |
|---|---|---|
| Sex | Female | 149 (94.9) |
| | Male | 8 (5.1) |
| Age (average ± standard deviation) | | 25.4 ± 3.3 |
| Licenses | Nurse | 157(100.0) |
| | Public health nurse | 31(19.7) |
| | Midwife | 1(0.6) |
| | Others | 1(0.6)* <br> * school nurse |
| Highest level of education in nursing (Number of years required for basic nursing education) | Upper secondary school major (5 years) | 8(5.1) |
| | Vocational school (3 years) | 84(53.5) |
| | Junior college (2 years) | 5(3.2) |
| | University (or college) (4 years) | 58(36.9) |
| | Graduate school | 1(0.6) |
| | N/A | 1(0.6) |
| Years of nursing experience | 3rd year | 74 (47.1) |
| | 4th year | 65 (41.4) |
| | 5th year | 18 (11.5) |

of the exploratory factor analysis were meaningful, although the sample size was small (Table 2). Factor 1 included "I consulted with my seniors about problems with teaching," "I asked my seniors to provide guidance to NGNs in my absence," "We encouraged each other as teachers for NGNs," and so on. Factor 1 was named "Experience of receiving cooperation from others," because it encompassed experiences of seeking guidance from senior nurses, as well as instances of mutual support among instructors. Factor 2 included "I told the NGNs that I could be their advisor," "I made time for the NGNs to talk about their concerns at work," and so on. Factor 2 was named "Experience of being there for the NGNs" because it included experiences of helping NGNs feel at ease at work. Factor 3 included "I felt good about my role as a NGN teacher," "I saw a positive change in the NGN through my guidance," and so on. Factor 3 was named "Experience of feeling positive about teaching" because it included successful experiences of teaching. Factor 4 included "I was troubled because I could not understand the NGNs' ideas," "I had a hard time communicating with the NGNs," and so on. Factor 4 was named "Experience of struggling with teaching" because it included experiences of distress related to the relationship with NGNs. Factor 5 included "I set/modified goals based on the NGNs' achievement status each time," "I checked the progress of knowledge and skills with the NGNs," and so on. Factor 5 was named "Experience of teaching according to level of achievement" because it included the experience of working closely with NGNs to guide them.

### Reliability verification

Cronbach's alpha coefficients for content consistency ranged from.849 to.923 for each factor and.924 overall (Table 3). The modified item-total correlations ranged from.255 to.744, and the Cronbach's alpha coefficients ranged from.918 to.925 after item deletion. The Cronbach's alpha coefficient was not significantly affected by the deletion of any item.

### Visualization results of factor structure

The chi-square value indicating the fit between the factors and the structure of each item was $\chi^2 (314) = 637.117$, $p < .001$, with CFI = .871 and RMSEA = .081 (Appendix). This was moderate when compared to the criteria for the structural validity of factor models.

To confirm criterion-related validity, we calculated the correlation coefficient between the total score of the 27 items remaining after exploratory factor analysis and the total score of the CNCSS. The Shapiro-Wilk test ($p = .007$) did not confirm the normality of the total score of the 27 items; however, the histogram and QQ plot results suggested that the scores were normally distributed. The Pearson correlation coefficient between the total score of all 27 items and the total score of CNCSS was.222 ($p < .01$).

## Discussion

### Sample adequacy

Of the study participants, 94.9% were female, and 5.1% were male. The percentage of male nurses is lower than that of female nurses worldwide, at 16% in Europe, 14% in the Americas, and 21% in Southeast Asia [24]. In comparison, the percentage of male nurses in Japan is lower, at 8.6% [25]; therefore, the gender ratios in this study generally reflect the sex distribution of nurses in Japan. Regarding age, although there are multiple routes to obtaining a nursing license in Japan, most respondents studied at a three-year vocational school or a four-year university, with those in their third to fifth year corresponding to ages24–26 (university students: 25–27 years). Therefore, the average age of the respondents in this survey was 25.5 years old, which is representative of those who have been engaged in nursing for three to five years. The sample size was 157. In factor analysis, the ratio of the number of participants to the number of factors should be 20:1 or higher; in the case of five factors, it should be 100 or higher [26]. The KMO measure of factor validity was.867, and Bartlett's test of sphericity was significant at $p < .001$. Although the sample size was small (n = 157), we believe that the results of the factor analysis conducted on this population are valid.

**Table 2. Exploratory factor analysis of the Newly Graduated Nurses Teaching Experience Scale (n = 157).**

| | | Factor | | | | |
|---|---|---|---|---|---|---|
| | | 1 | 2 | 3 | 4 | 5 |
| Factor 1<br>Experience of receiving cooperation from others<br>(Cronbach's α = .923) | 1 Seniors directly communicated with me about the NGNs' growth. | **.905** | −.216 | .077 | −.125 | .064 |
| | 2 Seniors sympathized with my experience teaching NGNs. | **.849** | .163 | −.078 | .027 | −.161 |
| | 3 Seniors asked me if I had any problems while teaching NGNs. | **.795** | −.041 | −.159 | −.049 | .049 |
| | 4 I received advice from seniors on how to instruct NGNs. | **.785** | .095 | −.009 | .014 | −.060 |
| | 5 I talked to seniors about my problems with teaching. | **.763** | .102 | −.090 | .054 | −.023 |
| | 6 I consulted with my seniors about specific guidance methods for NGNs. | **.754** | −.032 | .143 | .129 | −.067 |
| | 7 I asked my seniors to provide guidance to NGNs when I was not available. | **.601** | −.098 | .104 | .036 | .237 |
| | 8 Nurses with less experience in teaching newly graduated nurses encouraged each other. | **.477** | −.010 | .099 | −.045 | .256 |
| Factor 2<br>Experience of being there for the NGNs<br>(Cronbach's α = .854) | 9 I tried to create an atmosphere where NGNs felt comfortable talking to me about things other than work. | −.088 | **.845** | −.088 | −.192 | −.061 |
| | 10 I tried to empathize with NGNs' stories. | −.021 | **.699** | −.052 | .017 | −.196 |
| | 11 I told NGNs that I could be a consultant for them. | .030 | **.648** | −.032 | −.017 | .189 |
| | 12 I paid attention to the relationship between the NGNs and other staffs. | .052 | **.638** | −.015 | .185 | .010 |
| | 13 I ensured that the NGNs and I could openly express our thoughts to each other. | −.007 | **.602** | .225 | .007 | .023 |
| | 14 I felt that I was in charge of the NGNs and that I cared about the NGNs. | .031 | **.550** | .174 | −.073 | .147 |
| | 15 I made time for the NGNs to tell me what they were worried about in their work. | .080 | **.487** | −.020 | .009 | .272 |
| Factor 3<br>Experience of feeling positive about teaching<br>(Cronbach's α = .865) | 16 I felt that my perspective on education was broadened. | .079 | −.037 | **.896** | −.009 | −.088 |
| | 17 I felt glad to have been involved in the training NGNs. | .001 | .041 | **.831** | −.123 | −.026 |
| | 18 I gained confidence in education. | −.179 | −.125 | **.791** | −.015 | .113 |
| | 19 Through the NGNs' guidance, I came to recognize myself as I am now. | −.029 | .220 | **.648** | .179 | −.104 |
| | 20 I could see positive changes in NGNs through my guidance. | .101 | .003 | **.548** | −.041 | .008 |
| Factor 4<br>Experience of struggling with teaching<br>(Cronbach's α = .849) | 21 I was troubled because I could not understand NGNs' ideas. | −.001 | −.071 | −.030 | **.870** | −.021 |
| | 22 I had a difficult time communicating with NGNs. | −.123 | −.043 | −.020 | **.854** | .073 |
| | 23 I was troubled because I could not observe the NGNs' positive attitude toward learning. | .105 | .080 | −.110 | **.735** | −.023 |
| | 24 I was troubled by the teaching method due to differences from the basic education course. | .069 | −.040 | .113 | **.549** | .044 |

*(Continued)*

**Table 2.** (Continued)

| | | Factor | | | | |
|---|---|---|---|---|---|---|
| | | 1 | 2 | 3 | 4 | 5 |
| Factor 5<br>Experience of teaching according to level of achievement<br>(Cronbach's α = .856) | 25 Goals were set/modified each time based on the NGNs' achievement status. | .057 | −.049 | −.112 | −.021 | **1.036** |
| | 26 Progress of knowledge and skills was checked with the NGNs. | .045 | .083 | .000 | −.004 | **.681** |
| | 27 Discussed goals with the NGNs based on the educational program. | −.042 | −.044 | .168 | .141 | **.621** |
| Interfactor correlation | 1 | 1.000 | | | | |
| | 2 | .470 | 1.000 | | | |
| | 3 | .465 | .521 | 1.000 | | |
| | 4 | .491 | .052 | .056 | 1.000 | |
| | 5 | .650 | .418 | .530 | .350 | 1.000 |

Cumulative contribution ratio:59.572% Cronbach's α (all) : .924   Kaiser-Meyer-Olkin (KMO) measures:.867   Bartlett's sphericity test : p < .001.

Factor Extraction Method: Maximum Likelihood Method   Rotation Method: Promax Method with Kaiser Normalization.

## Factor structure and validity of NGNs' teaching experience scale

Exploratory factor analysis revealed a five-factor, 27-item scale structure, with each factor consisting of "Experience of receiving cooperation from others (Factor 1)," "Experience of being there for the NGNs (Factor 2)," "Experience of feeling positive about teaching (Factor 3)," "Experience of struggling with teaching (Factor 4)," and "Experience of teaching according to level of achievement (Factor 5)."

The experiences of "Receiving cooperation from others" included not only passive experiences such as "receiving advice on teaching NGNs from seniors" but also active experiences such as "consulting with seniors about teaching problems" and "consulting with seniors about specific teaching methods for NGNs." This implies that teaching NGNs should not be the responsibility of a single person; rather, it should involve nurses around the NGNs to support the nurse providing guidance as necessary. Additionally, it is necessary to build relationships among staff members who can consult each other for guidance. In addition, the experience of "Being there for the NGNs" included "letting NGNs know that I can be their advisor" and "making it easy for NGNs to talk about things other than work." It is important for teachers to be supportive and practical [27–29], and these extracted factors reinforce the results of previous studies.

The "Experience of teaching according to level of achievement" included direct guidance to NGNs, such as "setting/modifying goals based on the NGNs' achievement status" and "checking the progress of knowledge and skills with the NGNs." This means that even nurses with no experience in teaching NGNs demonstrate a teaching role that contributes to the NGNs' goals. Considering that the in-service education system is designed to enhance the nursing practice skills of NGNs, this experience is essential. The "Experience of feeling positive about teaching" included "I gained confidence in teaching" and "I was able to recognize myself through teaching NGNs." This indicates that even if respondents were unfamiliar with the experience of teaching NGNs, they felt positive changes as a result of their involvement in teaching NGNs. The positive experiences of teaching NGNs strongly influence the clinical teaching behavior of teaching nurses [30], suggesting that the teaching experiences of newcomers may be a trigger for improving the "teaching" skills of teaching nurses. On the other hand, the "Experience of struggling with teaching" includes the experience of struggling to "understand NGNs' ideas" and struggling with "communicating with NGNs," indicating that the experience of teaching NGNs is accompanied by challenges in relating to NGNs. An instructor's motivation and self-perception may change due to the actions they take to overcome challenges during teaching, which may be exhibited autonomously [31]. Hence, even experiences of struggling with teaching can be meaningful.

**Table 3. Reliability analysis of the Newly Graduated Nurses Teaching Experience Scale.**

| Factor | Items | Corrected Item Total Correlation | Cronbach's α coefficient if item is deleted | Cronbach's α (All:.924) |
|---|---|---|---|---|
| Factor 1 | 1 Seniors directly communicated with me about the NGNs' growth. | .680 | .919 | .923 |
| | 2 Seniors sympathized with my experience teaching NGNs. | .673 | .919 | |
| | 3 Seniors asked me if I had any problems while teaching NGNs. | .565 | .921 | |
| | 4 I received advice from seniors on how to instruct NGNs. | .698 | .919 | |
| | 5 I talked to seniors about my problems with teaching. | .669 | .919 | |
| | 6 I consulted with my seniors about specific guidance methods for NGNs. | .744 | .918 | |
| | 7 I asked my seniors to provide guidance to NGNs when I was not available. | .730 | .918 | |
| | 8 Nurses with less experience in teaching newly graduated nurses encouraged each other. | .641 | .920 | |
| Factor 2 | 9 I tried to create an atmosphere where NGNs felt comfortable talking to me about things other than work. | .255 | .925 | .854 |
| | 10 I tried to empathize with NGNs' stories. | .255 | .925 | |
| | 11 I told NGNs that I could be a consultant for them. | .562 | .921 | |
| | 12 I paid attention to the relationship between the NGNs and other staffs. | .553 | .921 | |
| | 13 I ensured that the NGNs and I could openly express our thoughts to each other. | .543 | .921 | |
| | 14 I felt that I was in charge of the NGNs and that I cared about the NGNs. | .573 | .921 | |
| | 15 I made time for the NGNs to tell me what they were worried about in their work. | .594 | .920 | |
| Factor 3 | 16 I felt that my perspective on education was broadened. | .550 | .921 | .865 |
| | 17 I felt glad to have been involved in the training NGNs. | .470 | .922 | |
| | 18 I gained confidence in education. | .351 | .924 | |
| | 19 Through the NGNs' guidance, I came to recognize myself as I am now. | .565 | .921 | |
| | 20 I could see positive changes in NGNs through my guidance. | .446 | .923 | |
| Factor 4 | 21 I was troubled because I could not understand NGNs' ideas. | .319 | .925 | .849 |
| | 22 I had a difficult time communicating with NGNs. | .307 | .925 | |
| | 23 I was troubled because I could not observe the NGNs' positive attitude toward learning. | .401 | .924 | |
| | 24 I was troubled by the teaching method due to differences from the basic education course. | .414 | .923 | |
| Factor 5 | 25 Goals were set/modified each time based on the NGNs' achievement status. | .700 | .919 | .856 |
| | 26 Progress of knowledge and skills was checked with the NGNs. | .612 | .920 | |
| | 27 Discussed goals with the NGNs based on the educational program. | .588 | .921 | |

The structural fit of each factor and item obtained from the exploratory factor analysis was examined: χ² (314) = 637.117, p<.001. However, the χ² value alone does not necessarily determine structural fit, as a good fit can exist even when it is rejected [32]. The CFI was.871, the RMSEA was.081. The CFI should be above.90 or.95, and the RMSEA should be below.05. For RMSEA, the range of.05 to.10 is difficult to determine, but it should not exceed.10. The CFI was slightly lower than.90, but the difference was minimal. Therefore, the results of the factor analysis were considered to have no major structural issues.

This is the first scale that measures the experiences of teaching NGNs. When the correlation between this scale and nursing practice ability was compared to a previous study [21], the correlation coefficient was.222, indicating a virtually non-existent correlation. The Self-Evaluation Scale of Preceptor Role Performance for New Graduate Nurses [18] used in Shiraki et al.'s study [21] measures "behaviors essential for preceptors to fulfill their role in teaching NGNs" and shows a correlation of.54 with nursing competence. Both scales measure the relationship between nurses providing guidance and NGNs, but the Self-Evaluation Scale of Preceptor Role Performance for New Graduate Nurses targets nurses with an average of 5.7 years of experience and 1.6 years in the preceptor role. As the clinical backgrounds of the nurses in this study differ from those of the nurses in the previous study, one possible reason for the lack of correlation with nursing practice ability was considered. However, further verification of criterion-related validity is needed in the future.

### Usefulness of the NGN's teaching experience scale

We believe that the NGNs' Teaching Experience Scale developed in this study will enable nursing managers to understand the extent to which nurses with less experience in teaching NGNs experience the process of working with them. It has been challenging for nursing managers to understand the specific experiences of less experienced nurses who are teaching NGNs during on-the-job training, even though they may observe them throughout the training process. Utilizing this scale for nurses with less experience in teaching NGNs can help nursing managers evaluate the challenges associated with teaching and how they relate to NGNs, providing a basis for designing support to improve the teaching skills of instructor nurses. It is possible to evaluate whether NGNs' education is progressing well based on their nursing practice and goal achievement status; however, by utilizing this scale, one can examine the situation from the perspectives of both the NGNs and the nurses supporting them, and consider measures for improving the instructional system for NGNs.

### Limitations

This study developed a scale to measure the extent to which nurses with three to five years of experience working in Japanese hospitals with more than 400 beds, who have never been responsible for teaching NGNs, experience teaching NGNs. However, it is necessary to test whether the scale can be adapted to hospitals with fewer than 400 beds, as NGNs may begin working at these facilities. Furthermore, although the sample size in this study met the minimum requirements, we conducted exploratory factor analysis and verified the structural consistency of the factors and items with a small sample size, which we consider a limitation of the research design. In addition, this study only included domestic nurses, further research is needed to expand the scope of hospital bed capacity and participant eligibility—including whether it could be applied to overseas hospitals—and to verify convergent validity with other measures. It is also reasonable to assume that some hospitals have changed their approach to on-the-job training before and after the COVID-19 pandemic, meaning that the nature of new employee guidance may have evolved in such facilities.

### Conclusions

The NGNs' Teaching Experience Scale comprised 27 items across 5 factors, consisting of "Experience of receiving cooperation from others," "Experience of being there for the NGNs," "Experience of feeling positive about teaching," "Experience of struggling with teaching," and "Experience of teaching according to level of achievement." The internal consistency, construct validity, and criterion-related validity of the scale were confirmed, allowing for the measurement of

the extent to which nurses with little experience teaching NGNs experienced teaching NGNs. The internal consistency and factor structure of this scale were confirmed; however, its correlation with nursing practice ability was low, and further verification regarding criterion-related validity is needed. To develop this scale into a usable instrument, future research must address the limitations identified in this study.

## Supporting information

**S1 Table. Summary of the participants (n = 157).**
(PDF)

**S2 Table. Exploratory factor analysis of the Newly Graduated Nurses Teaching Experience Scale (n = 157).**
(PDF)

**S3 Table. Reliability analysis of the Newly Graduated Nurses Teaching Experience Scale.**
(PDF)

**S4 Appendix. Visualization Results of Factor Structure.**
(TIF)

## Acknowledgments

We thank all the nurses who took the time out of their busy schedules to participate in this study.

The results of this study summarize part of the findings of a Master's thesis submitted to the Master's Program in the Division of Health Sciences, Osaka University Graduate School of Medicine.

## Author contributions

**Investigation:** Tomoki Doi.

**Methodology:** Tomoki Doi.

**Project administration:** Tomoki Doi.

**Supervision:** Yasuko Shimizu.

**Writing – original draft:** Tomoki Doi.

**Writing – review & editing:** Tomoki Doi.

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
