## [Decision Letter · Decision Letter 0]

9 Oct 2025

Dear Dr. Doi,

Thank you for submitting your manuscript to PLOS ONE. After careful consideration, we feel that it has merit but does not fully meet PLOS ONE’s publication criteria as it currently stands. Therefore, we invite you to submit a revised version of the manuscript that addresses the points raised during the review process.

ACADEMIC EDITOR: Experts in the field have reviewed your manuscript and you are expected to address their comments as early as possible. Thank you./>==============================

We look forward to receiving your revised manuscript.

Kind regards,

Olutosin Ademola Otekunrin

Academic Editor

PLOS ONE

2. In the online submission form, you indicated that [The data used in the analysis is owned by the researchers, but it can be provided upon request. The data was obtained from subjects who agreed to its use solely for the purposes of this study, and therefore it cannot be provided for use for other purposes for ethical reasons.].

3. Please remove your figure from within your manuscript file, leaving only the individual TIFF/EPS image files, uploaded separately. These will be automatically included in the reviewers’ PDF.

4. We notice that your supplementary figure is uploaded with the file type 'Figure'. Please amend the file type to 'Supporting Information'. Please ensure that each Supporting Information file has a legend listed in the manuscript after the references list.

Additional Editor Comments:

Experts in the field have reviewed your manuscript and you are expected to address their comments as early as possible. Thank you.

Reviewers' comments:

Reviewer's Responses to Questions

1. Is the manuscript technically sound, and do the data support the conclusions?

Reviewer #1: Yes

Reviewer #2: Partly

2. Has the statistical analysis been performed appropriately and rigorously?

Reviewer #1: Yes

Reviewer #2: No

3. Have the authors made all data underlying the findings in their manuscript fully available?

Reviewer #1: Yes

Reviewer #2: No

4. Is the manuscript presented in an intelligible fashion and written in standard English?

Reviewer #1: Yes

Reviewer #2: No

Reviewer #1: Review of the Manuscript: "Testing the Reliability and Validity of a Newly Graduated Nurses Teaching Experience Scale

Conceptual Clarity & Applicability

The focus of the study on assessing nurses' teaching experiences seems more appropriate for qualitative exploration. Identifying teaching experiences through qualitative methods would provide a more comprehensive understanding. Following this, the development of an instrument based on specific characteristics and conditions required for nurses to act as educators could have been more effective. As a result, I was unable to establish a clear connection with the instrument's practical application as presented in the current form of the paper.

Writing Quality

The manuscript would benefit from an improvement in its writing style. In some sections, the tone resembles a report rather than a scientific article. This can be adjusted to make the paper more formal and aligned with the expectations of scholarly writing.

Abstract

In the abstract, more information should be provided about the instrument, such as the number of initial items included. Additionally, mentioning the ethics committee approval in the abstract is unnecessary and should be omitted.

Content Validity

I noticed that there is no mention of content validity analysis (CVI & CVR). Including a thorough discussion of these measures would significantly strengthen the paper’s methodological rigor.

Software for CFA

Could you clarify which software was used for the Confirmatory Factor Analysis (CFA)? Providing this information will enhance the transparency of your methodological approach.

Sample Size and Limitations

While your sample size meets the minimum requirement, it would be beneficial to acknowledge a potential limitation. Since both Exploratory Factor Analysis (EFA) and CFA were performed on the same sample, it would be prudent to discuss this as a limitation in your study’s design.

Reviewer #2: The manuscript is valuable in its intent, but the conceptualization of the construct and the statistical analysis require revision. First of all, it is a major methodological error to perform EFA and CFA on the same sample. There is no mention of dividing the sample into two subsamples. When CFA is conducted on the same dataset used for EFA, CFA is no longer confirmatory analysis but only a circular verification. Moreover, on a sample of 157 people in the case of an instrument with 27 items, it is not advisable to perform CFA. The authors attempt to justify this by citing an author who suggests a ratio between the number of participants and the number of factors (lines 262-263). The 20:1 ratio is an outdated empirical rule applicable to EFA, not CFA. CFA is not based on the ratio of the number of participants per factor but on the complexity of the model and the degree of saturation. In CFA, the general guideline rule is 10 participants (and, according to some authors, even 25) per estimated parameter. Recent literature (e.g., Kline, 2016; Hair et al., 2022) offers much more demanding standards than Paul K (1993) proposes. It seems to me that this justification appears post hoc and seems intended to rationalize a questionable methodological decision, especially given that the authors acknowledge that the sample is small.

Conceptual inconsistency of the construct. Secondly, it is stated that it is about a Teaching Experience Scale for nurses with little teaching experience, but the dimensions identified do not describe the actual teaching experience but general social and emotional perceptions.

The authors state that they derived the items from previous literature but do not indicate which conceptual dimensions were taken over or modified.

The correlation with CNCSS (r=.222) is quite low, indicating very modest criterion validity. Although the authors show that it is significant, such a value explains less than 5% of the common variance.

In a measure validation process, it was expected that other forms of validity (convergent, discriminant, etc.) would also be achieved.

Only one comparable instrument is mentioned, the Self-evaluation Scale of Preceptor Role Performance for New

Graduate Nurses (although there are others—e.g., the Preceptor Self-Efficacy Scale and the Nursing Clinical Teaching Effectiveness Inventory), but no systematic comparison is made between the instrument itself and the Self-evaluation Scale of Preceptor Role Performance for New

Graduate Nurses.

Line 239 - CFI value of 0.871 does not meet the minimum standards. The value of CFI ≥ 0.90

Data collection procedures are also unclear. It is mentioned that administrators distributed the questionnaires, suggesting a paper-and-pencil format, but this must be clearly specified. It is important to also mention the environment in which the questionnaires were completed: at the hospital, at home, in working hours etc. The data are from 8 years ago (2017). In the context of post-pandemic changes, the generalizability of the data is low.

Table 2. - to be verified. includes a potential error: factor loading greater than 1.0

Minor observation: Line 167 - an interesting way of phrasing the significance threshold!!!

Major revision requested: increase the number of participants and create two subsamples. EFA to be done on one sample and CFA to be done on another sample.

Do you want your identity to be public for this peer review? For information about this choice, including consent withdrawal, please see our Privacy Policy

Reviewer #1: No

Reviewer #2: No

---

## [Author Response · Author response to Decision Letter 1]

9 Jan 2026

We sincerely thank the reviewers again for their constructive and insightful comments, which have greatly strengthened the quality of our manuscript. Detailed revisions addressing each comment have been uploaded as files via the “Attach Files” section. We hope that the revisions adequately address all concerns, and we respectfully submit the revised manuscript for your reconsideration.

---

## [Editor Report · Decision Letter 1]

3 Feb 2026

Testing the reliability and validity of a newly graduated nurses' teaching experience scale

PONE-D-25-32129R1

Dear Dr. Doi,

We’re pleased to inform you that your manuscript has been judged scientifically suitable for publication and will be formally accepted for publication once it meets all outstanding technical requirements.

Kind regards,

Olutosin Ademola Otekunrin

Academic Editor

PLOS One
---

## [Editor Report · Acceptance letter]

PONE-D-25-32129R1

PLOS One

Dear Dr. Doi,

I'm pleased to inform you that your manuscript has been deemed suitable for publication in PLOS One. Congratulations! Your manuscript is now being handed over to our production team.

Kind regards,

on behalf of

Dr. Olutosin Ademola Otekunrin

Academic Editor

PLOS One